evolution, palaeontology

competition, macroevolution, biotic interactions, fossil biodiversity, Granger causality, bryozoa

**Authors for correspondence:**
Scott Lidgard
e-mail: slidgard@fieldmuseum.org
Lee Hsiang Liow
e-mail: l.h.liow@nhm.uio.no

# When fossil clades 'compete': local dominance, global diversification dynamics and causation

Scott Lidgard[1], Emanuela Di Martino[2], Kamil Zágoršek[4] and Lee Hsiang Liow[2,3]

[1]Negaunee Integrative Research Center, Field Museum, 1400 S. Lake Shore Drive, Chicago, IL 60605 USA
[2]Natural History Museum, and [3]Centre for Ecological and Evolutionary Synthesis, Department of Biosciences, University of Oslo, Oslo, Norway
[4]Department of Geography, Technical University of Liberec, Studentská 2, CZ-461 Liberec, Czech Republic

SL, 0000-0002-0446-4705; LHL, 0000-0002-3732-6069

Examining the supposition that local-scale competition drives macroevolutionary patterns has become a familiar goal in fossil biodiversity studies. However, it is an elusive goal, hampered by inadequate confirmation of ecological equivalence and interactive processes between clades, patchy sampling, few comparative analyses of local species assemblages over long geological intervals, and a dearth of appropriate statistical tools. We address these concerns by reevaluating one of the classic examples of clade displacement in the fossil record, in which cheilostome bryozoans surpass the once dominant cyclostomes. Here, we analyse a newly expanded and vetted compilation of 40 190 fossil species occurrences to estimate cheilostome and cyclostome patterns of species proportions within assemblages, global genus richness and genus origination and extinction rates while accounting for sampling. Comparison of time-series models using linear stochastic differential equations suggests that interclade genus origination and extinction rates are causally linked to each other in a complex feedback relationship rather than by simple correlations or unidirectional relationships, and that these rates are not causally linked to changing within-assemblage proportions of cheilostome versus cyclostome species.

## 1. Introduction

Time after time during life's history, a major clade of organisms seems to be displaced by another with presumably similar ecological characteristics [1]. This recurring pattern was once explained by assertions that competition gradually favoured a better-adapted group over its rival [2]. Now, a wave of studies is developing analytical frameworks aimed at narrowing the gaps between ecological interactions, changing taxonomic dominance and deep time clade dynamics [3–7]. Besides accounting for sampling and other biases [8], understanding how clade interaction and changing taxonomic dominance work depends on timing, rates and processes in a hierarchy of geographical, temporal and taxonomic levels [9–12]. Inferring causation from this understanding is largely but not entirely a matter of explanatory reduction, focusing on whether entities and processes at a higher level can be explained to some degree by entities and processes at a lower level [13–15].

The challenges are formidable for a study of causal linkages in apparent clade displacement. In studies of biotic interactions between fossil members of different clades, the ecological equivalence of participants is seldom established conclusively and empirical evidence of competitive mechanisms is rarely preserved, let alone quantified [16,17]. Bryozoans are singular exceptions.

Cyclostome and cheilostome bryozoans are allied phylogenetically [18], comparable physiologically and ecologically [19] and co-occur on the same benthic substrates [20,21]. Among encrusting bryozoan colonies competing for space, cheilostomes are routinely the overgrowth 'victors' [22–24]. The fossil history of cheilostome taxonomic richness surpassing cyclostome richness locally within assemblages at the species level and globally at the genus level is a canonical example of clade displacement [2,20,21,25–27]; but a stubbornly unresolved evolutionary question is whether or not local taxonomic dominance and global diversification dynamics are causally linked. Moreover, causal linkages between cheilostome and cyclostome global diversity dynamics have never been quantified.

Clade displacement studies typically measure changes in genus richness for each competing clade as proxies for changing patterns at lower scales [16], or (more recently) pair such empirical data with model-based phylogenetic comparative methods [7,11,12]. Yet studies seldom document concurrent long-term species proportions among individual fossil assemblages for competing clades [20,28,29]. Additionally, inadequate taxonomic synonymization may bias empirical fossil diversity patterns [30]. Appropriate estimates of global diversity patterns are needed to account for sampling artefacts, requiring a sufficient set of occurrences through time to establish diversification rates and their uncertainty. The focus of clade displacement at a macroevolutionary scale is on changing global rates of origination and extinction of taxa—is there evidence of predictability and/or temporal correlation of rates between clades, or with changing local species proportions? Formally comparing models of these patterns and processes provides a basis for causal inferences [10].

Here, we present data and analyses designed to meet these challenges. We compile and present data cataloguing cyclostome and cheilostome bryozoans over the past 150 Myr, identifying each genus/species, place of occurrence and chronostratigraphic age (electronic supplementary material). An enlarged set of 40 190 fossil species occurrences more than doubles the data in previous bryozoan studies [20,25–27,31] and far surpasses species level documentation for other examples of clade displacement [4,28,32]. We first analyse changing cyclostome–cheilostome species proportions within local fossil assemblages. Global genus richness patterns for the two clades are then compared through 33 geological stages, incorporating taxonomic revisions owing to synonymizing and accounting for heterogeneous sampling. We then estimate the underlying genus origination and extinction rates for the two clades. Capture–mark–recapture (CMR) methods [3] are employed in our genus-level analyses to model origination and extinctions simultaneously with sampling rates and hence to account for incomplete sampling, both absent in prior fossil bryozoan biodiversity studies.

Finally, we investigate potential correlations and causal connections within and between cheilostomes and cyclostomes by comparing time series of genus origination and extinction rates as well as a time series of within-assemblage species proportions. We use the statistical concept of Granger causality [33] that originated in econometrics and is now employed across many disciplines as a probabilistic method for investigating *predictions* between different time series [34,35] wherein 'cause' in one time series statistically informs 'consequence' in another. In Granger's original work [33, p. 430] comparing two time series $X_t$ and $Y_t$, 'If some other series $Y_t$ contains information in past terms that helps in the prediction of $X_t$ and if this information is contained in no other series used in the predictor, then $Y_t$ said to cause $X_t$'. Thus, a potential cause may be constituted by information in one time series that both precedes and predicts its effect in a different time series.

Causality in this study is derived from processes modelled from observations of fossil occurrences compared over entire time series and is agnostic to underlying complexity (e.g. ecological interactions of organisms that are neither directly observed nor explicitly modelled). Statistical comparison affords a perspective of relative support for dynamic temporal correlations, lack of relationship and either unidirectional or 'feedback' Granger causality between pairs of models. Using linear stochastic differential equations (SDEs) [3,36], we evaluate strengths of correlation or Granger causal relationships among eight pairwise time-series analyses involving global genus-level origination and extinction rates and within-assemblage species proportions for the two clades. These relationships inform two main hypotheses: (i) that increased cheilostome genus origination rates dampened cyclostome origination rates or increased their extinction rates, and (ii) that temporal patterns of change in local assemblage proportions of cheilostome and cyclostome species can detectably impact and thus contribute to the explanation of global genus diversification rates.

## 2. Methods

### (a) Data

We restrict our expanded compilation of published fossil bryozoan occurrences to the Tithonian (152.1–145 Ma) through to the Holocene (0.01 Ma-present). We impose an arbitrary cut-off to exclude references published earlier than the year 1920, as determining locations of occurrence, their geographical extent, reliable geological ages and verifying taxonomic determinations (and synonymizations) is exceedingly difficult for older publications. We manually compiled two databases, *FosLocal* and *Age-Only*, facilitating analyses at local and global levels (text-mined versions are also discussed in the electronic supplementary material). Both databases include three basic elements: the identity of each genus/species, place of occurrence and chronostratigraphic age of occurrence.

The FosLocal database contains cheilostomes and cyclostomes reported from a single locality and stratigraphic horizon, nearly all identified to species level. The geographical area of a locality varies among references, but corresponds roughly to a local fossil assemblage for purposes of within-assemblage species richness comparisons. If a publication reports species from more than one stratigraphic horizon at a given geographical location, each horizon is retained as a separate locality entry. This database contains 1695 locality entries and 36 155 occurrences and is used for our analyses of within-assemblage fossil species proportions.

The Age-Only database contains genus/species names that do not necessarily come from a single locality, but are constrained geographically to a localized region (i.e. state, country or depositional basin). It contains 308 locality or regional entries and 4418 genus/species occurrences. The combined database merges data from the Age-Only and the FosLocal database and is used for our analyses of fossil global genus richness. See the electronic supplementary material for details on geological age assignment and taxonomic vetting.

### (b) Within-assemblage species richness and proportions

For all localities ($n = 1695$) in the FosLocal database, species counts of cheilostomes and cyclostomes are plotted in figures

*Proc. R. Soc. B* **288**: 20211632

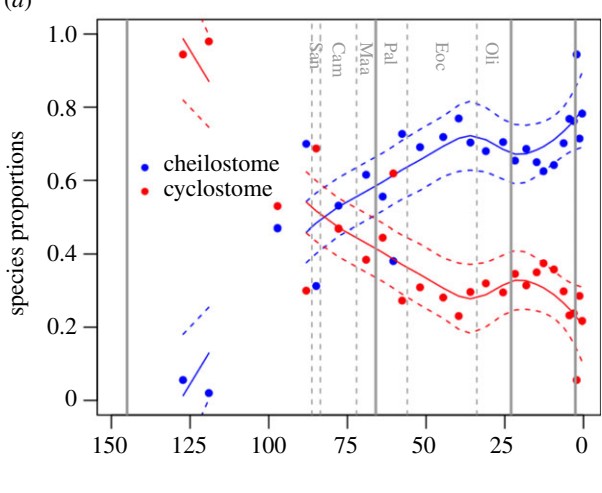

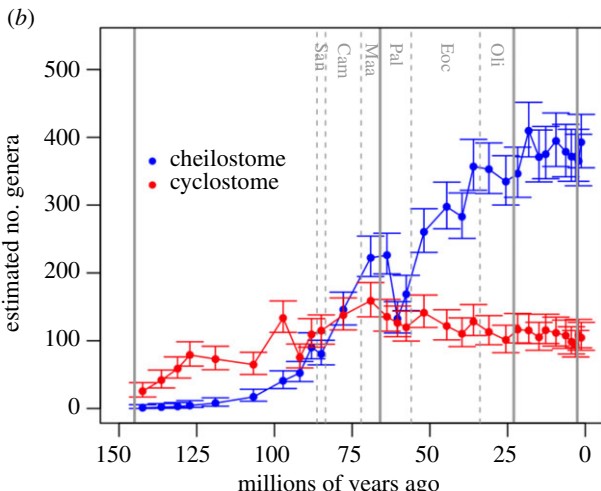

**Figure 1.** Changing taxonomic dominance at local species and global genus levels. (*a*) Patterns of temporal change for within-assemblage species proportions of cheilostome species (blue lines) and cyclostome species (red lines) estimated by nonparametric LOESS regression with 95% confidence intervals. Dots show the mean within-assemblage proportions of cheilostome species (blue) and cyclostome species (red) calculated for each stratigraphic stage. Only assemblages including both cyclostomes and cheilostomes and at least 15 species are included in fitting the curves (sensitivity analyses in the electronic supplementary material, figures S2 and S3). (*b*) Genus richness estimated using a CMR Jolly–Seber model and synonymized occurrences with 95% confidence intervals. For reference with the text, geological periods (Jurassic, Cretaceous, Palaeogene, Neogene, Quaternary) are separated by solid vertical lines and key Cretaceous stages/ages (San = Santonian, Cam = Campanian, Maa = Maastrichtian) and Palaeogene epochs (Pal = Palaeocene, Eoc = Eocene, Oli = Oligocene) are separated by dashed vertical lines. (Online version in colour.)

at the midpoints of their assemblage age ranges (electronic supplementary material, figure S1) using stages listed in the International Chronostratigraphic Chart of the International Commission on Stratigraphy (electronic supplementary material, table S1).

Proportions of cheilostome versus cyclostome species are calculated using subsets of the localities in the FosLocal database in which the author(s) report the presence of both cheilostomes and cyclostomes ($n = 975$). In figure 1*a*, we estimate the timing of 'cross-over' in higher species proportions from cyclostomes to cheilostomes by applying nonparametric locally estimated scatterplot smoothing (LOESS) regression, with the default degree of smoothing $\alpha = 0.75$, using the predict.loess function in base R v. 4.1.0 [37]. We analyse sensitivity of within-assemblage percentage trends to minimum species numbers per assemblage at

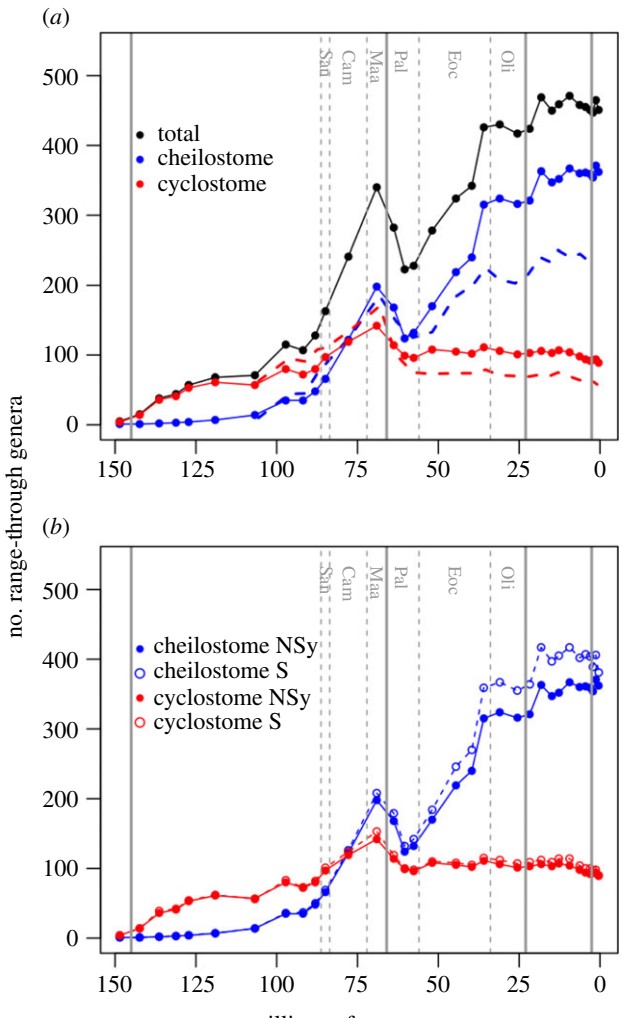

**Figure 2.** Genus range-through curves. (*a*) This panel plots our new, synonymized genus data (solid circles and lines) for cheilostomes (blue) and cyclostomes (red) separately and together (black). The dashed lines (blue = cheilostomes, red = cyclostomes) show data from McKinney & Taylor [30] which is the basis for even the most recent review [18]. (*b*) This panel repeats the synonymized genus data from (*a*) for cheilostome and cyclostome genera separately but also plots the non-synonymized data (open circles and dotted lines) for comparison. Vertical lines and abbreviations as in figure 1. (Online version in colour.)

three levels (electronic supplementary material, figure S2), retaining localities with at least 10 species (765 localities), 15 species (630 localities) and 20 species (531 localities). To further validate these within-assemblage patterns using LOESS regressions, we fit separate cubic splines through proportions of cheilostome and cyclostome species (electronic supplementary material, figure S3) using the smooth.spline function in R.

### (c) Global genus range-through richness in comparison to previous compilations

To compare our expanded fossil genus biodiversity data with older compilations based only on reported first and last stratigraphic occurrences [18,20,25–27,30], we follow such publications in assuming that genera are extant only from the time of their first observation to the time of their last observation and plot range-through genus richness (figure 2; electronic supplementary material, figure S4). We also compare the raw, non-synonymized data and the synonymized data, where in the latter there were 694 cheilostome genera and 261 cyclostome genera in the interval encompassing the Tithonian and the Holocene.

## (d) Global genus richness estimation

The above range-through approach assumes that a genus cannot have existed before its first observation or after its last observation, and that the only genera that can be unobserved are ones that have observations in two or more time intervals in the dataset. Additionally, confidence intervals for genus richness cannot be generated in a straightforward way when using range-through tabulations. These assumptions are relaxed with a Jolly–Seber model [38,39], reviewed by Pollock *et al.* [40], which also uses information from the non-observation of genera both within and outside of the range-through intervals. The Jolly–Seber model is an open population model in the CMR literature (reviewed by King [41]) that estimates 'population size' (genus richness in our analyses), 'survival rates' (the complement of extinction in our analyses) and 'birth numbers' (number of genus originations in our analyses). An important limiting assumption is that all genera have the same probability of being sampled. For instance, we expect very lightly calcified genera to be preserved and sampled only rarely; our conclusions are probably reflecting the preservation of moderately to well-calcified genera. Other assumptions, including short sampling intervals relative to the time over which survival is estimated and independence of genera are not thought to bias estimates [40,42,43]. We implement the Jolly–Seber model using the JS.direct function in the openCR R package [44]. To estimate 95% confidence intervals for the number of genera estimated, we assume a Poisson sampling model (figure 1b; electronic supplementary material, figure S5).

We also compare Jolly–Seber estimates with estimates using PyRate [45], an approach that makes different assumptions about sampling, as a means of substantiating these genus biodiversity patterns (electronic supplementary material, figure S7). In PyRate, the probability of each genus is assumed to change through its duration but estimation is conditioned on at least one observation of a genus (electronic supplementary material).

## (e) Genus origination and extinction estimation

For genus origination and extinction rates, we use another CMR model, the Pradel seniority model [46], as described in previous palaeontological studies [47–50]. It combines forward and reverse-time modelling to examine both 'survival' and 'seniority,' whose complements translate to extinction and origination for genus-level data. The Pradel seniority framework is flexible in that extinction, origination and sampling probabilities can be time-varying or may include covariates (e.g. cheilostomes and cyclostomes could be constrained to have different estimates in the same model). We employ fully time-varying models for cheilostomes and cyclostomes using our combined dataset as well as a text-mined-dataset [51] using the openCR.fit function while specifying 'Pradelg' [44] (figure 3; electronic supplementary material, figure S6).

The origination and extinction rate estimates from the Pradel model are transition probabilities across temporal boundaries. Sampling probabilities, however, are associated with the time intervals themselves. All estimated probabilities are transformed into instantaneous rates by assuming a Poisson model [3], as our 33 time intervals are unequal in duration. We also substantiate these rates by comparing CMR estimates with those from PyRate analyses [45] (electronic supplementary material, figures S8 and S9).

## (f) Link model analyses using linear stochastic differential equations

We quantify relative support for whether cyclostome and cheilostome genus origination and extinction time series may have no detectable relationship to one another, be correlated with or casually linked to one another and/or have such relationships with a time series of changes in species proportions in local species

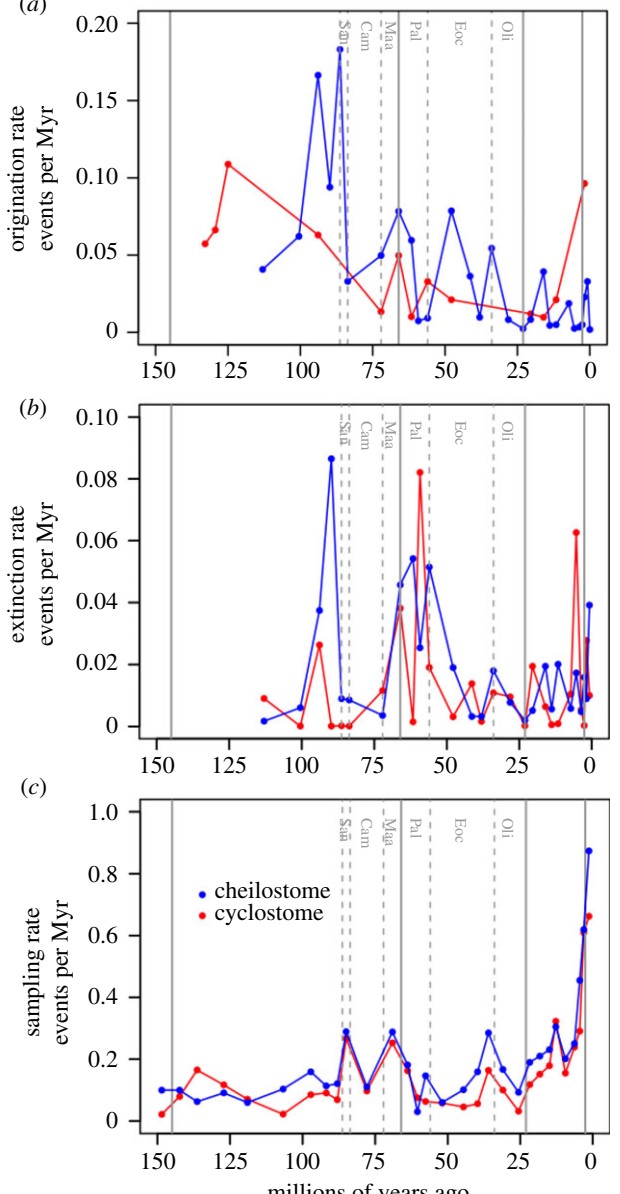

**Figure 3.** Genus origination, extinction and sampling rates. Each panel shows the estimates from fully time-varying Pradel seniority models run separately for cheilostome genera (blue) and cyclostome genera (red). Only relatively well-constrained estimates are plotted as dots (confidence intervals shown in the electronic supplementary material, figure S6). Lines joining the dotted estimates are for visual aid only. (*a*) Instantaneous origination rates, (*b*) instantaneous extinction rates, and (*c*) sampling rates. Sampling rates are within-stage; origination and extinction rates are for stage boundaries. Vertical lines and abbreviations as in figure 1. (Online version in colour.)

assemblages. In part because time-series approaches commonly used in palaeontology, such as first differencing, are prone to false inferences [3,10], we employ a time-series approach based on linear SDEs [52] to investigate causality. This system of linear SDEs explicitly models temporal correlations versus Granger causality [33,53] among time series while relaxing the requirement of sampling at temporally equidistant points, and embracing uncertainties associated with the time-series estimates [54,55]. In addition to comparing null, correlation and causal models, we also provide parameter estimates from each model, including estimates of the strengths of correlations and Granger causal relationships among time series.

A linear SDE can be written as

$$dX_1(t) = -\alpha_1(X_1(t) - \mu_1)\,dt + \sigma_1 dB_1(t), \tag{2.1}$$

**Table 1.** Causal links among genus diversification rates and within-assemblage species proportions. (Pairs of time-series variables (columns) are evaluated using five different models of relationships (rows). Numbers in cells show the per cent support (Bayesian posterior probabilities) for different models (parameter estimates shown in the electronic supplementary material, tables S2 and S3). Grey cells indicate the best model. Abbreviations: cheilostome (cheil) and cyclostome (cycl) genus origination (orig) and extinction (ext) rates; proportions of cyclostome species to combined species in faunal assemblages (proportion).)

| models | time series | | | |
|---|---|---|---|---|
| | 1. cheil orig 2. cycl orig | 1. cheil ext 2. cycl ext | 1. cheil orig 2. cycl ext | 1. cheil ext 2. cycl orig |
| A. no relationship between time series | 41.8 | 11.4 | 49.2 | 24.1 |
| B. 1st time-series drives 2nd | 10.5 | 22.9 | 11.4 | 9.8 |
| C. 2nd time-series drives 1st | 20.4 | 13.4 | 14.5 | 24.4 |
| D. temporal feedback between time series | 17.9 | 44.0 | 14.6 | 31.2 |
| E. correlation between time series | 9.4 | 8.3 | 10.2 | 10.5 |
| | 1. cheil orig 2. proportion | 1. cheil ext 2. proportion | 1. cycl orig 2. proportion | 1. cycl ext 2. proportion |
| A. no relationship between time series | 34.9 | 73 | 44.4 | 38.2 |
| B. 1st time-series drives 2nd | 18.5 | 11.8 | 14.4 | 12.1 |
| C. 2nd time-series drives 1st | 14.7 | 10.6 | 15 | 16.9 |
| D. temporal feedback between time series | 25.2 | 0 | 18.4 | 24.9 |
| E. correlation between time series | 6.6 | 4.6 | 7.8 | 7.9 |

where $X_1(t)$ is the process (time series) of interest (e.g. cheilostome origination rates) and where the first part of the right side of (equation (2.1)) is an ordinary differential equation and the second part is a stochastic component. $\alpha$, $\mu$ and $\sigma$ represent the strength of the attracting force towards an average value, the value of this average, and the intensity of random fluctuations, respectively. To model a pair of times series, we add an equation:

$$dX_2(t) = -\alpha_2(X_2(t) - \mu_2)\,dt + \sigma_2(1 - \rho^2)^{0.5}dB_2(t) + \rho\sigma_2 dB_1(t),$$
(2.2)

where for example, $X_1(t)$ represents cheilostome origination rates and $X_2(t)$ is cyclostome origination rates. Here equations (2.1) and (2.2) are linked via $dB_1(t)$ such that $\rho$ represents the strength of the correlation between the two origination rates. The temporal troughs and peaks of the two processes will be strongly correlated if $\rho$ is high, even though $X_1(t)$ does not influence the other $X_2(t)$ and vice versa.

We can also express one process as a function of the other such that a change in one process occurs before a change in the other process, i.e. Granger causality occurs.

In the following, $X_2(t)$ is controlled by $X_1(t)$ in the sense that when $X_1(t)$ changes (in the deterministic term), $X_2(t)$ must follow:

$$dX_2(t) = -\alpha_2(X_2(t) - \mu_2 - \beta[X_1(t) - \mu_1])\,dt + \sigma_2 dB_2(t)\,dt. \quad (2.3)$$

The Granger causal relationship between $X_2(t)$ and $X_1(t)$ is here summarized by $\beta$. We use the term 'link model' when referring to models describing correlation or causal relationships.

We implement link model analyses using the R package layzeranalyzer [36]. We log transform the four extinction and origination rate time series to conform to the normality requirement in our linear SDE tool kit. To the species proportion time series, we add a small value (0.0001) to allow the log transformation for normalization. We apply pairwise comparisons among the five time series rather than a multi-time-series comparison, as our data series are too short for meaningful multi-series comparisons [3,36]. Bayesian posterior model probabilities are presented in table 1 and Bayes factor is used for further model comparisons [56]. We also present parameter estimates and their uncertainties (table 2; electronic supplementary material, tables S2 and S3) for time-series pairs (table 1) for which the null hypothesis is clearly rejected. $\alpha$ is re-parameterized where we report half-lives $t_{1/2} = \log(2)/\alpha$ for the processes involved, facilitating ease of interpretation.

## 3. Results

Species richness of cyclostomes and cheilostomes among local fossil assemblages is highly variable. The assemblages are not sampled uniformly through time, the general tendency being greater density toward the Recent. Both cheilostome species and total species per assemblage generally increase through time despite significant declines in the Palaeocene and Oligocene and lesser fluctuations through the Neogene (electronic supplementary material, figure S1). Both nonparametric LOESS regressions and cubic splines fitted to the assemblage data support an inference that the average within-assemblage proportion of cheilostome species probably started to exceed that of cyclostomes roughly 85–75 Ma (end-Santonian or early to mid-Campanian; figure 1a; electronic supplementary material, figures S2 and S3).

Jolly–Seber model estimates take sampling into account [3] and are based on our synonymized, combined dataset. They suggest a 'crossover' to higher global genus richness of cheilostomes roughly 77–69 Ma (later Campanian or early Maastrichtian; figure 1b; electronic supplementary material, figure S5). The Jolly–Seber model estimates are similar to alternative richness estimation with the same dataset using PyRate [45] (electronic supplementary material, figure S7).

A comparison of range-through genus richness with the most comprehensive published compilation [18,30] shows that our compilation has contributed a substantial amount of new data. Higher global genus richness for both clades is

**Table 2.** Parameter estimates from the best models. (Estimated values are presented for mean, median and lower and upper bounds of the best models that are not null models from table 1. The values of $\mu$ are the means (logged rate), $t_{1/2}$ (in millions of years) are the half-lives, $\sigma$ are the stationary variances and $\beta$ are the strengths of the Granger causal link between the two time series.)

| | mean | median | lower bound (95%) | upper bound (95%) |
|---|---|---|---|---|
| *relationship between cheilostome and cyclostome extinction model D* | | | | |
| $\mu$.cheilostome.extinction | −2.126 | −2.088 | −3.641 | −0.605 |
| $t_{1/2}$.cheilostome.extinction | 30.344 | 14.116 | 0.719 | 136.945 |
| $\sigma$.cheilostome.extinction | 0.288 | 0.294 | 0.002 | 0.623 |
| $\mu$.cyclostome.extinction | −2.459 | −2.527 | −3.728 | −0.939 |
| $t_{1/2}$.cyclostome.extinction | 13.710 | 3.474 | 0.460 | 77.983 |
| $\sigma$.cyclostome.extinction | 0.195 | 0.152 | 0.002 | 0.647 |
| $\beta$.cheilostome.extinction.to.cyclostome.extinction | 0.522 | 0.572 | −0.345 | 1.212 |
| $\beta$.cyclostome.extinction.to.cheilostome.extinction | 0.326 | 0.442 | −1.017 | 1.315 |
| *relationship between cheilostome extinction and cyclostome origination model D* | | | | |
| $\mu$.cheilostome.extinction | −2.482 | −2.493 | −3.795 | −1.070 |
| $t_{1/2}$.cheilostome.extinction | 14.64 | 6.600 | 0.641 | 80.398 |
| $\sigma$.cheilostome.extinction | 0.197 | 0.168 | 0.002 | 0.577 |
| $\mu$.cyclostome.origination | −2.026 | −2.036 | −3.269 | −0.472 |
| $t_{1/2}$.cyclostome.origination | 29.319 | 18.183 | 0.708 | 121.260 |
| $\sigma$.cyclostome.origination | 0.275 | 0.252 | 0.009 | 0.658 |
| $\beta$.cheilostome.extinction.to.cyclostome.origination | 0.275 | 0.303 | −0.915 | 1.185 |
| $\beta$.cyclostome.origination.to.cheilostome.extinction | 0.563 | 0.637 | −0.773 | 1.314 |

increasingly apparent from the Eocene onwards (figure 2*a*). Our synonymization of genera in published literature shows that raw non-synonymized range-through genus richness prominently inflates cheilostome genus richness from about the middle Eocene onwards (figure 2*b*).

Despite fewer Mesozoic than Cenozoic data, analyses indicate that during the Cretaceous, cheilostome genus origination rates significantly exceed cyclostome ones (figure 3*a*; electronic supplementary material, figure S6), and cheilostome extinction rates are mostly lower (figure 3*b*; electronic supplementary material, figure S6). For many Cretaceous and Cenozoic stages, higher cheilostome origination rates contrast sharply with cyclostome rates, as do net differences between origination and extinction (similar PYRATE estimates shown in the electronic supplementary material, figures S8 and S9). Sampling rates are similar for cyclostomes and cheilostomes, even though they are estimated separately (figure 3*c*). Furthermore, Cretaceous cheilostome genus origination rates surpass cyclostome rates well before cheilostomes attained local species dominance (electronic supplementary material, figures S2, S3 and S6).

From earlier prominent studies on clade displacement [18,22,23,57], we hypothesized that cheilostome origination would dampen cyclostome origination (upper portion of table 1, time series column 1, model B) and/or increase cyclostome extinction (column 3, model B), and that cheilostome extinction would facilitate cyclostome origination (column 4, model B). However, cheilostome origination rates have decidedly poorer detectable relationships to cyclostome origination or extinction rates, given the higher weights for the model of no relationship between time series (columns 1, 3, model A). Between rates of cheilostome and cyclostome extinction, a

bi-directional feedback model has the highest model weight (column 2, model D) where the Bayes factor is 44/11.4 = 3.9, indicating moderate support for model D compared with the null model of no relationship (model A). Parameter estimates for this model are shown in table 2 (top portion).

While a bi-directional feedback model also had the highest weight in the pairwise comparison of cheilostome extinction and cyclostome origination (upper portion of table 1, column 4, model D), the Bayes factor (comparing model D with the null model, A) is only 1.3 in this case, indicating weak support for the alternative model D, given our data. The precise nature of feedback is ambiguous, unsurprising as the time series themselves are uncertain and short. Considering that the mean parameter estimate (table 2, top portion) $\beta$.cheilostome.extinction.to.cyclostome.extinction = 0.522 (credibility interval of −0.345 to 1.212) is more positive than negative, one acceptable inference is that high cheilostome extinction rates could be related to high extinction rates for cyclostomes, and vice versa. It is plausible that at the same time, high cheilostome extinction rates also induce higher cyclostome origination rates (table 2 bottom, $\beta$.cheilostome.extinction.to.cyclostome.origination mean parameter estimate = 0.275) and higher origination rates in cyclostomes in turn drive higher cheilostome extinction rates. However, in the latter comparison model support is lower and $\beta$ estimates have larger uncertainty bounds (table 2). These results differ from simple active displacement of cyclostomes by cheilostomes as advocated in previous bryozoan studies. Instead, cheilostome extinction has a perceptibly stronger effect on cyclostome extinction than vice versa (table 2 top portion, $\beta$.cheilostome.extinction.to.cyclostome.extinction = 0.522 versus $\beta$.cyclostome.extinction.to.cheilostome.extinction 0.326). In addition, one group's

partial disappearance (cheilostomes) could have influenced another's increased origination (cyclostomes), as also reported previously for bivalves and brachiopods [3].

We also hypothesized that global cheilostome and/or cyclostome genus origination and/or extinction rates could be driven by temporal changes in within-assemblage proportions of species belonging to each clade. If this were the case, we would expect lower per cent support for model A (no relationship between time series) than for models B, C, D, E, or a combination thereof in the lower portion of table 1. Instead, the weights for the model of no relationship between time series are uniformly highest across all the pairwise time-series comparisons (table 1 lower portion, columns 1–4, model A). Changing local species proportions and global genus rates do not even share the same dynamics, given the uniformly low weights for time-series correlation (model E).

## 4. Discussion

Our compilation and analyses bring to light several matters important to past and current large-scale fossil biodiversity studies—not ours alone. It seems intuitively obvious that global genus-level evolutionary dynamics must be linked in some ways to what is happening at lower scales: species, ecological populations, local communities. However, despite our best efforts with a very large dataset and an up-to-date modelling approach, links between changes in species taxonomic dominance within fossil assemblages and diversity dynamics estimated for genera in the two clades were not supported. In seeking quantifiable and testable hypotheses, one needs to determine what can actually be measured consistently and with sufficiently numerous observations in the fossil record. A long-established palaeontological consensus has settled on using genera as the most viable data for estimating global biodiversity and diversification rates [58], yet there are many mismatches at smaller local scales, where individual organisms and populations really interact [16].

Fossil evidence of organisms in one clade competing directly with those in another is scarce or simply absent for most metazoan taxa. We do have empirical evidence for encrusting cheilostomes' expected superiority in fossil and Recent overgrowth of cyclostomes [22–24], but we lack sufficient quantification of overgrowth interactions between cheilostomes and cyclostomes in our assemblage data to analyse whether overgrowth interactions are detectably linked to diversification rates in our Granger causal analyses. While proportional species richness is an important parameter in its own right, it is only a proxy for relative abundance, and probably even less so for dominance in overgrowth interactions. Furthermore, other life-history attributes (e.g. reproductive or recruitment strategies) may facilitate the persistence of members of the 'subordinate' clade by avoiding overgrowth encounters [24–27]. The competing cyclostomes persisted locally without sustained decline in global richness as cheilostomes continued diversifying through the Cenozoic [20,21,30]. Perhaps more importantly, there are multiple ways that patterns at a higher level could be realized through underlying mechanisms. This makes it more complicated to say that one particular ecological mechanism (spatial competition via overgrowth of individual colonies in different species) is responsible for a macroevolutionary pattern (genus richness or diversification rates). To be clear, it is entirely plausible that competitive outcomes *could*

influence species composition within and among assemblages, and—indirectly—genus diversification rates. However, sufficient quantification of direct competitive interactions among fossil organisms within local assemblages for this and other clade displacement studies is not yet within reach.

Another possible metric of interest at local scales is ecological abundance of putatively competing clades; how are time series of changing relative abundance within assemblages related to global diversity dynamics? Long-term fossil ecological dominance in terms of relative abundance (numbers of individuals or biomass) has seldom been evaluated owing to scarce data, and then mostly as coarse assessments. The few exceptions from micropalaeontological and palynological studies [59,60] usually span shorter geological intervals. Abundance is even more problematic for colonial organisms like bryozoans, which are often preserved as fragmented skeletons of an unknown number of colonies. Alternatively, long-term dominance in terms of relative species richness within assemblages frequently matches that of relative abundance [61,62], while only allowing that these measures can be decoupled sporadically [28,31]. The former metric is what we employed; it can be measured consistently and with sufficient recurrence for our analyses.

Both this and other large-scale fossil biodiversity analyses confront artefacts of sampling, with potential biases that have been a focus of studies for at least a half-century [63–66]. This variety of artefacts and their possible effects caution us against taking unprocessed 'raw' sample occurrence data at face value in view of differing durations of successive time intervals, incomplete preservation, availability and temporal uniformity of suitable sedimentary rocks and of sample occurrences, monographic and systematic biases and more. Concurrent with this history, an array of models has been developed to account for incomplete sampling and other biases in the raw fossil data [8,66], with PyRate and CMR methods among the most recent [3–7,43,48]. These approaches have proved effective in helping to mitigate many of the effects of sampling biases by statistically processing and abstracting from the temporal incompleteness and heterogeneity of the raw data [66,67], enabling palaeobiologists to better estimate the underlying biological signal of changes in biodiversity and origination and extinction rates. Still, there remain artefacts that are not easily overcome. Virtually all broad fossil biodiversity studies reflect the preponderance of work by past and present researchers in more developed countries and associated influence from greater numbers of surveyed locations that today reside in the Northern Hemisphere [68,69]. What we and others present are 'snapshots' based on what is currently known and then what subset of this knowledge is included in a given analysis.

Greater resolution of real underlying biological signals from data and subsequent models would surely come from more extensive efforts at taxonomic revision, finer stratigraphic resolution, more uniform temporal sampling and greater inclusion of the sparsely represented geographical regions including the palaeotropics and Southern Hemisphere [70,71]. As an example, consider that most of the spectacular increase in known bryozoan genus and species richness from the Campanian stage (83.6–72.1 Ma) through to the Danian stage (66–61.6 Ma) is confined to the Northern European Chalk Sea, with many hundreds of species still undescribed [30,61,72,73]. This province of shallow epicontinental pelagic carbonate ooze is perhaps unique in the Phanerozoic. The Cretaceous/Palaeogene extinction abruptly reduced bryozoan

species and genus richness here, and other Maastrichtian hold-overs died off at the end of the Danian (figure 1; electronic supplementary material, figures S1; S2) with the final disappearance of the Chalk Sea [21,73]. If most of the known end-Cretaceous cheilostomes evolved and later perished here, what characterized the remainder of the global fauna during this regional Late Cretaceous radiation, the pulsed extinction through the Palaeocene, and the subsequent massive diversification of cheilostomes later in the Palaeogene [73]? If more complete global sampling was available, biodiversity patterns would probably be altered, but how much would estimated global diversification dynamics differ (figure 3; electronic supplementary material, figure S6)? While shorter-duration excursions in rates or proportions are 'real', they say little about statistical predictability in a Granger causal sense [10,35]. One efficient approach for expanding the data for all such studies might use natural language processing methods to survey publications that are not yet incorporated into existing compilations [51,74] (electronic supplementary material, figures S10, S11 and S12).

Temporal and local geographical patterns of reputedly competing clades' co-occurrence alone do not signify interactions directly [35]. Among the most salient features of our analyses is the quantification of causal interactions, correlations, or the deficiency of both of these among time series. The fossil genus data are drawn from an extraordinarily large and dispersed sample set, most genera have broader geographical distributions and longer geological durations than do species and are thus more likely to be detected, and the CMR analyses take sampling irregularities into account. Given these factors and the relative weights for models compared in the upper portion of table 1, there is compelling evidence for complex bi-directional causal feedback between cheilostome global genus extinction rates and cyclostome origination and extinction, respectively. On the other hand, unidirectional causal relationships between global genus diversification rates of the two clades have less support.

As we indicated above, in fossil clade displacement one would intuitively anticipate some relationship (e.g. potential drivers) between lower-level patterns and processes and higher-level macroevolutionary dynamics. However, correlative or causal relationships are comparatively undetectable between time series for any of the global genus diversification rates and within-assemblage species proportions of the two clades (lower portion of table 1). Conceivable rationales underlying this lack of significant relationships are only speculative at present. It is possible that despite our use of the same extensive sample set, we are still unable to detect such relationships. Alternatively, time series other than those we modelled might also inform us about such relationships. Future comparative analyses could incorporate additional quantified time series of potential drivers—more extensive fossil documentation of competitive overgrowth interactions, abiotic Earth system variables such as palaeotemperature, area of continental flooding surfaces or oscillations in seawater carbonate chemistry [23,75,76]. We might also consider factors that have been associated qualitatively with cheilostome diversification, such as the phylogenetic expansion of key innovations [21,77,78] or particular environmental and ecological changes [30,79].

One more matter is the recognition that sampling of the fossil record can never be complete at local or global scales. We accept a recent view of palaeontological data models offered by Bokulich [66], that data models should be understood as representations, and that the fidelity of a data model in capturing the signal of interest is a matter of degree. In general, these data models will not be entirely free of bias or error; the expectation is that the models can still function adequately for a particular inquiry or as evidence for a hypothesis. Bokulich quoted palaeontologist Benton et al. [80, p. 63], 'We suggest that palaeontologists, like other scientists, should accept that their data are patchy and incomplete, and use appropriate methods to deal with this issue in each analysis. All that matters is whether the data are adequate for a designated study or not'. Looking forward, we should recognize that as shown in this study, datasets and data models evolve and can be refined over time. It seems likely to us that the accumulation of more detailed regional biodiversity studies at finer temporal scales, as 'case studies' ancillary to large-scale studies [67], will be especially valuable in refining the overall representation of competing clades. Cheilostomes and cyclostomes, like bivalves and brachiopods [3] were not merely 'ships that pass in the night' [81], but fuller understanding of the extent to which time-series histories of local species abundances and overgrowth interactions influenced macroevolutionary outcomes awaits future data compilation and analyses.

Data accessibility. All data and code for analyses are available from the Dryad Digital Repository: https://doi.org/10.5061/dryad.zpc866t6s [82].

Authors' contributions. S.L.: conceptualization, data curation, formal analysis, investigation, methodology, project administration, supervision, validation, visualization, writing—original draft, writing—review and editing; E.D.M.: data curation, writing—review and editing; K.Z.: data curation, writing—review and editing; L.H.L.: conceptualization, data curation, formal analysis, funding acquisition, investigation, methodology, project administration, software, supervision, validation, visualization, writing—original draft, writing—review and editing. All authors gave final approval for publication and agreed to be held accountable for the work performed therein.

Competing interests. We declare we have no competing interests.

Funding. The European Research Council supported this project under the European Unions's Horizon 2020 research and innovation programme (grant agreement no. 724324 to L.H.L.).

Acknowledgements. We thank D. Silvestro for advice on PYRATE, T. Reitan for advice on causal analysis, P. E. Bock and D. P. Gordon for sharing taxonomic lists, C. Schuette for assistance compiling data, and E. Håkansson and A. C. Love for comments.

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
