## [Peer Review File · Proceedings of the Royal Society B: Biological Sciences]

Review History

RSPB-2021-0637.R0 (Original submission)

Review form: Reviewer 1

Recommendation

Accept with minor revision (please list in comments)

Please rate the overall quality of this paper:

Good

Is it accessible?

Yes

Is it clear?

Yes

Is it adequate?

Yes

Do you have any concerns about statistical analyses in this paper? If so, please specify them explicitly in your report.

No

It is a condition of publication that authors make their supporting data, code and materials available - either as supplementary material or hosted in an external repository. Please rate, if applicable, the supporting data on the following criteria.

Comments to the Author

The paper presents a substantially enlarged dataset of species occurrences for fossil bryozoans, and then subjects this step change in data abundance to contemporary analytical procedures such as Grainger Causality and PyRate biodiversity projections. The central take-home message is that the origination and extinction dynamics link across the two focal clades, but not to relative abundance of the species. The authors conclude that we should reconsider the routes by which "bottom-up" competition regulates higher level biodiversity dynamics.

Generally, this is an interesting read that makes a number of valid points. I want to raise a couple of thoughts that the paper provoked.

The concluding statement (above, 380-381) implies that we have evidence to reject the "bottom-up" organisation of diversity within these two bryozoan clades. But, which of the models in Table 1 would argue the opposite? I don't disagree with the concluding statement and thrust of the argument, but am not sure you signpost directly where the test that rejects bottom-up structuring is. The comparison could plausibly relate to the comparison between Tables 1 and 2, but I don't think that is raised in the current Discussion and do think it would necessitate a brief reprise of the relative vs absolute abundance debates.

Tables 1 and 2 present % support for the various models, but are discussed without reference to uncertainty - the support for model D discussed 316-326 is, at 31.2%, not much more than either C (24.4%) or the null A (24.1%). I'm not advocating that the numbers are wrong, but that you should acknowledge the uncertainty in the %s alongside the descriptions of the consequences of the "uncertain and rather short" time series (319).

While discussing uncertainty, can we see confidence intervals on the splines - eyeballing the 95% CIs in Fig. 2b suggests that the divergence becomes clearly established later than the point estimate for the crossover point.

Table 2 needs some additional formatting to make it easier to read - you don't need to use μ , dt , σ in the table and can replace with the words from the legend, and do also explain what aspects of Grainger causality like "characteristic time" mean for those of us less familiar with the method (the description in the methods section is great; thank you!)

Finally, the pyRate reference is almost apologetic. I don't see how it adds much given you use the range-through numbers and the figure is only in the supplement. Either "go big" and use the PyRate uncertainties in the same panel as the range-through numbers or "go home" and drop it.

75 - insert only between while and allowing (if I understood the sentence correctly)

77 - need a reference for the generification

99 - is modelling the right word here? accounting for?

143 - what does "each one" refer to? Species, location or occurrence?

147 - how many does "some" mean?

162 - cite R version number

Review form: Reviewer 2

Recommendation

Major revision is needed (please make suggestions in comments)

Scientific importance: Is the manuscript an original and important contribution to its field?

Excellent

General interest: Is the paper of sufficient general interest?

Excellent

Quality of the paper: Is the overall quality of the paper suitable?

Acceptable

Is the length of the paper justified?

Yes

Should the paper be seen by a specialist statistical reviewer?

No

Do you have any concerns about statistical analyses in this paper? If so, please specify them explicitly in your report.

Yes

It is a condition of publication that authors make their supporting data, code and materials available - either as supplementary material or hosted in an external repository. Please rate, if applicable, the supporting data on the following criteria.

Is it accessible?

Yes

Is it clear?

No

Is it adequate?

No

Do you have any ethical concerns with this paper?

No

Comments to the Author

See attached PDF. (See Appendix A)

Decision letter (RSPB-2021-0637.R0)

17-May-2021

Dear Dr Lidgard:

I am writing to inform you that your manuscript RSPB-2021-0637 entitled "When fossil clades "compete": Local dominance, global diversification dynamics, and causation" has, in its current form, been rejected for publication in Proceedings B.

This action has been taken on the advice of referees, who have recommended that substantial revisions are necessary. With this in mind we would be happy to consider a resubmission, provided the comments of the referees are fully addressed. However please note that this is not a provisional acceptance.

The two reviewers found the study interesting and very ambitious but both had reservations or questions about the analyses and other substantive aspects of the paper, in quite some technical detail but all of them truly constructive critiques that seem like they could be addressed.

Both reviewers noted that improvements to the shared data/code are imperative-- better organization is needed and update the readme file with better manuscript identification information (e.g., full author list) as well as explanation of that organization.

Sincerely,
Dr John Hutchinson, Editor
mailto: proceedingsb@royalsociety.org

Reviewer(s)' Comments to Author:

Referee: 1

Comments to the Author(s)

The paper presents a substantially enlarged dataset of species occurrences for fossil bryozoans, and then subjects this step change in data abundance to contemporary analytical procedures such as Grainger Causality and PyRate biodiversity projections. The central take-home message is that the origination and extinction dynamics link across the two focal clades, but not to relative abundance of the species. The authors conclude that we should reconsider the routes by which "bottom-up" competition regulates higher level biodiversity dynamics.

Generally, this is an interesting read that makes a number of valid points. I want to raise a couple of thoughts that the paper provoked.

The concluding statement (above, 380-381) implies that we have evidence to reject the "bottom-up" organisation of diversity within these two bryozoan clades. But, which of the models in Table 1 would argue the opposite? I don't disagree with the concluding statement and thrust of the argument, but am not sure you signpost directly where the test that rejects bottom-up structuring is. The comparison could plausibly relate to the comparison between Tables 1 and 2, but I don't

think that is raised in the current Discussion and do think it would necessitate a brief reprise of the relative vs absolute abundance debates.

Tables 1 and 2 present % support for the various models, but are discussed without reference to uncertainty - the support for model D discussed 316-326 is, at 31.2%, not much more than either C (24.4%) or the null A (24.1%). I'm not advocating that the numbers are wrong, but that you should acknowledge the uncertainty in the %s alongside the descriptions of the consequences of the "uncertain and rather short" time series (319).

While discussing uncertainty, can we see confidence intervals on the splines - eyeballing the 95% CIs in Fig. 2b suggests that the divergence becomes clearly established later than the point estimate for the crossover point.

Table 2 needs some additional formatting to make it easier to read - you don't need to use μ , dt , σ in the table and can replace with the words from the legend, and do also explain what aspects of Grainger causality like "characteristic time" mean for those of us less familiar with the method (the description in the methods section is great; thank you!)

Finally, the pyRate reference is almost apologetic. I don't see how it adds much given you use the range-through numbers and the figure is only in the supplement. Either "go big" and use the PyRate uncertainties in the same panel as the range-through numbers or "go home" and drop it.

75 - insert only between while and allowing (if I understood the sentence correctly)

77 - need a reference for the generification

99 - is modelling the right word here? accounting for?

143 - what does "each one" refer to? Species, location or occurrence?

147 - how many does "some" mean?

162 - cite R version number

Referee: 2

Comments to the Author(s)

See attached PDF

Author's Response to Decision Letter for (RSPB-2021-0637.R0)

See Appendix B.

RSPB-2021-1632.R0

Review form: Reviewer 2

Recommendation

Accept with minor revision (please list in comments)

Scientific importance: Is the manuscript an original and important contribution to its field?

Excellent

General interest: Is the paper of sufficient general interest?

Excellent

Quality of the paper: Is the overall quality of the paper suitable?

Excellent

Is the length of the paper justified?

Yes

Should the paper be seen by a specialist statistical reviewer?

No

Do you have any concerns about statistical analyses in this paper? If so, please specify them explicitly in your report.

No

It is a condition of publication that authors make their supporting data, code and materials available - either as supplementary material or hosted in an external repository. Please rate, if applicable, the supporting data on the following criteria.

Is it accessible?

Yes

Is it clear?

Yes

Is it adequate?

Yes

Do you have any ethical concerns with this paper?

No

Comments to the Author

The authors have done a brilliant job of responding to my questions and suggestions, especially in the introduction, methods and results. Changes to the introduction make it much better suited to the study. The code is also now much more user friendly. (Just for the clarity, the suggestion to create sub-directories is based on the standard programming practice of putting data and scripts in different folders.)

The discussion has also been updated to include more discussion about incomplete sampling. My only outstanding criticism on this front is that, while interesting, the contents of the discussion don't currently link very well back to the authors' results. My suggestion is to improve the connection between the arguments & issues presented in the discussion and the results. I would also suggest remove the conclusions section because I don't think it's necessary (and especially wouldn't be needed if the discussion focused more on the results). Overall the paper makes a very valuable contribution to this area of research, and a slightly improved discussion would make it even better.

Minor comments

l82 "to model sampling rates" - do you mean to model the origination, extinction and sampling processes?

l104 "cheilostome genus origination rates would dampen cyclostome origination rates" - do you mean "did" instead of "would"?

l140 "4.1.0 (R Core Team 2019)" R version 4.1.0 wasn't available in 2019 but came out in 2021

1401 suggest citing the following: <https://eartharxiv.org/repository/view/2472/>. Recent work by Roger Close also seems relevant here.

From the supplement:

- "Extinction and origination rates from PyRate analyses are not directly comparable with those from the Pradel model. The latter are transition rates from one time interval to the next, while those from PyRate plotted in figs. S8 and S9 are averages within time intervals" - I could be wrong, but I don't think this means they can't be compared, since we typically interpret them in the same way, i.e. they play the same functional role in hypothesis testing and have been compared in simulation studies.
- "The two classes of models also incorporate different assumptions." - please also make it clear that the methods use different data.
- "We do not compare preservation (sampling) rates for the two models as they are very different in both assumptions and structure" - suggest replacing "structure" with "and the underlying data used" (or similar).
- "the uncertainty stems from the estimation of start and end points of taxon lifespans instead" - and presumably the stochasticity of the underlying birth-death process?

Decision letter (RSPB-2021-1632.R0)

23-Aug-2021

Dear Dr Lidgard

I am pleased to inform you that your manuscript RSPB-2021-1632 entitled "When fossil clades "compete": Local dominance, global diversification dynamics, and causation" has been accepted for publication in Proceedings B. Congratulations!!

The referee(s) have recommended publication, but also suggest some minor revisions to your manuscript. Therefore, I invite you to respond to the referee(s)' comments and revise your manuscript. Because the schedule for publication is very tight, it is a condition of publication that you submit the revised version of your manuscript within 7 days. If you do not think you will be able to meet this date please let us know.

The requested changes are mainly about linking the Results and Discussion more strongly, which should be very achievable and helpful.

[http://datadryad.org/submit?journalID=RSPB&manu=\(Document not available\)](http://datadryad.org/submit?journalID=RSPB&manu=(Document%20not%20available)) which will take you to your unique entry in the Dryad repository. If you have already submitted your data to dryad you can make any necessary revisions to your dataset by following the above link. Please see <https://royalsociety.org/journals/ethics-policies/data-sharing-mining/> for more details.

Sincerely,
Dr John Hutchinson
Editor
mailto: proceedingsb@royalsociety.org

Author's Response to Decision Letter for (RSPB-2021-1632.R0)

See Appendix C.

Decision letter (RSPB-2021-1632.R1)

26-Aug-2021

Dear Dr Lidgard

I am pleased to inform you that your manuscript entitled "When fossil clades "compete": Local dominance, global diversification dynamics, and causation" has been accepted for publication in Proceedings B.

Data Accessibility section

Open Access

Paper charges

Sincerely,

Appendix A

In this study, the authors explore the long-term co-evolutionary dynamics of two groups at local and global scales (cyclostome and cheilostomes bryozoans). They apply a new model-based approach to an incredible dataset. The dataset is almost unparalleled in terms of quality of sampling and reflects an enormous amount of compilation work. Thus, the paper has the potential to be very impactful.

I have a few comments on the methodology (described below) but my main criticism is that the way the paper is currently written doesn't serve the findings of the study well.

The paper starts out very boldly - the authors introduce the importance of inter-clade interactions and the abstract states "inter-clade dynamics are causally linked to each other". However, my impression after reading the results (I304-I340) is that the results are much less clear than this statement implies. Further, the finding that "no relationship between any of the origination and extinction rates and species proportions represented in local assemblages" requires more discussion. The patterns observed at a global scale must to *some* degree be linked to what's happening at lower spatial scales. The authors state that this results "defies the notion that diversification rates are driven by local representation" - but couldn't it be that we simply can't detect the relationship based on available data?

Overall the role of sampling is underplayed - it isn't really mentioned until the very short discussion. Sampling can never be truly "global" due to incomplete sampling and their modelling does not take this into account. This leads me to wonder whether biased estimates of global metrics are insufficient to be able to detect a causal relationship between assemblage level and global patterns. To me, this is one of the most interesting aspects of the results - since, if we can not establish a relationship between local and global scale dynamics based on a dataset of this quality, when would we ever be able to do this?

The structure of the paper is a bit unconventional - it has a very lengthy introduction that describes competitive replacement in detail and relatively small and unsatisfying discussion. The discussion doesn't really come back to competitive replacement or provide a biological explanation for what could be driving the patterns in cyclostomes and cheilostomes.

The concept of Granger is difficult to grasp from the introduction. Some of this might be better suited to the methods and given its importance to the results and that most readers will not be familiar with this, I suggest improving this description - perhaps by moving some of the supp explanation of this to the main text. PyRate could also be described in a bit more detail in the main text, to enable the reader to better understand the differences between the two modeling approaches (more comments on this below).

I suggest the authors revisit the structure of the manuscript, focus more on the role of sampling and on how this may affect the results that they obtain in the study. The analyses behind the study were quite well performed, however, the discussion section does not do justice to the results. Please consider expanding it. I also recommend the authors to provide more details regarding the methods in the main text. If the authors improved the text, I think it would make a more accessible and cohesive manuscript on a very important topic.

Methods

CMR vs PyRate: The comparison between PyRate versus CMR methods is a bit misleading. From the supplement: “PyRate also estimates origination and extinction rates given that taxa are sampled at least once, but capture recapture models relax this assumption, hence also contributing to larger (but more realistic) uncertainties.” While it is true that CMR relaxes this assumption, the two models can not actually be applied to the same data -- CMR uses sampled-in-bin (presence / absence) data while PyRate uses occurrence data. The assumption that taxa are sampled at least once does not explain the difference in uncertainties. Warnock et al. 2021 compare the birth-death process underlying PyRate to one that assumes incomplete species sampling (the fossilised birth-death process) and show that assuming complete has an adverse impact on accuracy - however, it does not have a discernible impact on uncertainty.

It could be the use of more data (i.e. occurrence data) leads to increased precision but a related issue is that PyRate requires exact fossil ages. Because CMR uses sampled-in-bin data, fossil age uncertainty is already accounted for. The developers of PyRate recommend sampling fossil ages from their known interval of uncertainty, repeating the analyses (e.g. 100 times) and constructing credible intervals based on the entire range of output. Ideally, the authors would apply this approach in this paper.

There's a bit of ambiguity about how the fossil ages are handled - “the combined successive stages are used to bracket an absolute age estimate” - does this refer to age brackets that span more than one interval? Can you clarify how ages that do only span one geological stage are handled? The authors mentioned that they relied on the stratigraphic placement given by the authors but this would still span a range, even if smaller. How were absolute ages calculated in this case? In addition, assigning absolute ages may introduce additional uncertainties in the analyses, how were these considered? The timescale used in the study also isn't specified in the main text. Perhaps you could add a table with age / stage info to the supp.

I also don't fully understand the following statement: “the uncertainty in origination and extinction rates stem directly from the distribution of genera known to be extant but otherwise unsampled” - how exactly does “the distribution of genera known to be extant but otherwise unsampled” lead to estimates uncertainty? Birth-death processes also lead to estimates of uncertainty (whether or not they account for incomplete sampling). Finally, the authors note that variation in sampling is not accounted for using CMR (although I believe it can be) but this can be accounted for using PyRate.

Spline fitting: The authors explored the impact of different degrees of freedom on the fit of cubic splines to their data but the authors did not mention how they decided which fit was chosen as the best fit. Was it based just on a visual inspection or did the user specifically use a model selection method? The R functions they use actually output information that can be used to select the best model - is there a reason the authors chose not to do this? Or if they did, it would be good to mention this.

Code: The code associated with this paper is poorly organised - it would not be straightforward for someone else to use. The authors should tidy it up, create subdirectories

for the data, scripts and output as well as include clear instructions for other users (i.e. by adding a README).

Figures

Suggest adding the official geological time scale to the figures to the plots so it's easier to switch back and forth between the figures and the text.

Minor

L94, "far surpasses species level documentation for other examples of clade displacement."
- Please mention some examples

I120, "The possibility of multiple realization" - I really have no idea what this means. This whole paragraph is difficult to follow.

I36, "facile assertions" - this seems a bit patronizing - the models in this paper also make simplifying assumptions!

I177, I wouldn't say naturally because this has an ambiguous statistical meaning - perhaps say "in a straightforward way".

I193, "in a wholly different manner" - this is very vague and I would actually say that the approaches are very similar. They both assume a Poisson process for origination, extinction and sampling, for instance.

I244, 245, should the number of time series be the same?

I286, "For many Cretaceous and Cenozoic stages, higher cheilostome origination rates contrast sharply with cyclostome rates" - what about the large spike at the end?

I304, "Surprisingly, cheilostome origination rates have no detectable relationship to cyclostome origination or extinction rates" is this result surprising, given the patterns in Fig 4a?

L320, "Considering the parameter estimates and Bayes factors between models" - be more explicit here.

19 July 2021

re: re-submission of previous reference number **RSPB-2021-0637**

Responses to referees comments:

Lidgard, et al. - "When fossil clades "compete": Local dominance, global diversification dynamics, and causation."

Dear Prof. Hutchinson and referees:

We appreciate that both referees recognized the effort involved in our compilation and analyses, and in general have not disagreed with the thrust of our conclusions. Here we respond to substantive comments made by both referees, but first resolve a contrast that arises between the two reviews with respect to our inclusion of PyRate as a secondary complement to our primary fossil biodiversification estimator, capture-mark-recapture (CMR). As R1 put it, "PyRate reference is almost apologetic ... or "go home" and drop it." On the other hand, R2 focused their comments on our secondary analyses, i.e. PyRate, and suggested increasing the number of iterations. Comparing these two different methods or other alternative methods was NOT the aim of this study. We provided PyRate estimates only because PyRate has sometimes been used as an alternative to previous (other) methods for estimating diversification rates.

In this revision we focus on CMR-based analyses in the main text, following one of the alternatives suggested by R1, in order to maintain the emphasis on the most novel aspects of our study there (i.e. the downstream causal analyses, results in Tables 1 and 2). But we also follow the recommendation of R2 and ran the PyRate analyses with 100 iterations to more fully account for age uncertainties of fossil observations in our revisions (new plots are presented in supplementary materials Fig. S7, S8, S9, equivalent to the old plots, but with 100 iterations). We reference PyRate as a means of comparing CMR estimates in the main text and present those comparisons in the revised supplementary material section 2.5 "Genus sampling, richness and rate estimates using PyRate." The general biodiversity trends revealed by CMR and PyRate are similar, as they were in the original MS. We note again that these methods differ in assumptions and also how confidence intervals are constructed. For instance, the CMR model we used (Pradel) also explicitly *includes* making inferences on genera that have never been sampled in diversification estimates, while PyRate does not. We based our main analyses on CMR for several reasons beyond those described in the MS: i) we know it and understand it better as Liow is directly familiar with its application in paleobiology, ii) our Granger causal analyses based on stochastic differential equations are designed (by Liow and colleagues) to deal with the types of uncertainties estimated in CMR analyses, iii) the uncertainties in PyRate and CMR analyses are very different in nature (noted in part by R2, e.g., "Warnock et al. 2021 compare the birth-death process underlying PyRate to one that assumes incomplete species sampling (the fossilised birth-death process) and show that assuming complete [sampling, i.e. PyRate, this added by us] has an adverse impact on accuracy." We also revise the abstract, shorten the introduction (removing original Fig. 1 in order to limit distraction from our principal analyses), and expand our results and discussion sections in the main text, especially to explain further the results in Tables 1 and 2 so readers can more thoroughly understand these novel time series analyses.

Line numbers listed below refer to the revised version of the MS (though we show the original line numbers in quoting comments from R1 and R2). *Our responses are in italics.*

Referee: 1

Comments to the Author(s) The paper presents a substantially enlarged dataset of species occurrences for fossil bryozoans, and then subjects this step change in data abundance to contemporary analytical procedures such as Grainger Causality and PyRate biodiversity projections. The central take-home message is that the origination and extinction dynamics link across the two focal clades, but not to relative abundance of the species. The authors conclude that we should reconsider the routes by which "bottom-up" competition regulates higher level biodiversity dynamics. Generally, this is an interesting read that makes a number of valid points. I want to raise a couple of thoughts that the paper provoked.

The concluding statement (above, 380-381) implies that we have evidence to reject the "bottom-up" organisation of diversity within these two bryozoan clades. But, which of the models in Table 1 would argue the opposite? I don't disagree with the concluding statement and thrust of the argument, but am not sure you signpost directly where the test that rejects bottom-up structuring is. The comparison could plausibly relate to the comparison between Tables 1 and 2, but I don't think that is raised in the current Discussion and do think it would necessitate a brief reprise of the relative vs absolute abundance debates.

*We removed the terms "bottom-up" and "top-down" from the conclusion. These terms appear to be more laden with inadvertent meaning than we intended, particularly by conjuring up relative abundance. What we analyze in the MS is local relative species RICHNESS, not ecological abundance as measured by the numbers of individuals or biomass in each species, summed for cheilostomes and for cyclostomes. (It may be that we've misunderstood the intended meaning of R1's comment "but not to relative abundance of the species.") We tried to indicate in the original text that adequate data on relative abundance is simply unavailable here and for virtually all other published large-scale analyses of fossil biodiversity. Abundance is even more problematic for colonial organisms like bryozoans, which are often preserved as fragmented skeletons of an unknown number of colonies. We are reluctant to delve into past "relative vs absolute abundance debates" (or local vs global richness, proportional vs absolute richness)—that's not what is new in this study. Those arguments are already well-known in the fossil biodiversity literature, including Lidgard's initial study of clade interactions in bryozoans, which is cited in this MS (Lidgard S, McKinney FK, Taylor PD. 1993 Competition, clade replacement, and a history of cyclostome and cheilostome bryozoan diversity. *Paleobiology* **19**, 352–371). [line 84-107]. In an effort to avoid misunderstanding, we added more explanation about Grainger causality, removed the original Fig. 1, shortened the introduction and relocated some of the text on relative abundance and other matters. Revised versions of that introductory text are now presented in the discussion alongside brief consideration of 'uncertainty' and statements on local versus global perspectives of dominance in taxonomic diversity.*

[line 338-345] Each of the models specifying different causal links in Table 1 should be judged by relative percent support. We revised the relevant paragraph to address R1's question "which of the models in Table 1 would argue the opposite?" This paragraph now reads: "We also hypothesized that global cheilostome and/or cyclostome genus origination and/or extinction rates could be driven by temporal changes in within-assemblage proportions of species belonging to each clade. If this were the case, we would expect lower percent support for model A (no relation between time series) than for models B, C, D, E, or a combination thereof in the lower portion of Table 1. Instead, the weights for the model of no relation between time series are uniformly highest across all the pairwise time series comparisons (columns 1-4, model A).

Changing local species proportions and global genus rates do not even share the same dynamics, given the uniformly low weights for time series correlation (model E)."

Tables 1 and 2 present % support for the various models, but are discussed without reference to uncertainty - the support for model D discussed 316-326 is, at 31.2%, not much more than either C (24.4%) or the null A (24.1%). I'm not advocating that the numbers are wrong, but that you should acknowledge the uncertainty in the %s alongside the descriptions of the consequences of the "uncertain and rather short" time series (319).

[lines 226-232] To clarify the numbers in the tables, we now state that Table 1 presents Bayesian posterior probabilities and cite Jefferys (1998) in the text and in section 2.6. "Causal analyses using model comparison and linear SDEs" in the supplementary materials. These changes provide some context to help the reader unfamiliar with Bayesian analyses. Basically, the rule of thumb (proposed by Jeffreys and used widely) is that a Bayes Factor (ratio of Bayesian posterior probabilities for models being compared) of 1 points to no evidence of one model being better than the other, 1-3 with "anecdotal evidence" and 3-10 with moderate evidence. Ref: Jeffreys H. 1998 Theory of Probability. Third Edition. Oxford: Oxford University Press.

[lines 294-336] The Bayesian posterior probabilities in Table 1 (and their ratios, see supplementary material section 2.6), combined with the parameter estimates and their credibility intervals allow us to make the quantitative inferences that we did. We inserted clarifications in text surrounding Tables 1 and 2 to aid interpretation. We also note that there were some copy-paste errors from our analyses output in the submitted Table 2 which are now corrected.

While discussing uncertainty, can we see confidence intervals on the splines - eyeballing the 95% CIs in Fig. 2b suggests that the divergence becomes clearly established later than the point estimate for the crossover point.

[line 398]. Original Fig. 2 is revised as new Fig. 1, showing separate nonparametric LOESS regression curves with 95% confidence intervals for within-assembly proportions of cheilostomes and cyclostomes (2a), and genus biodiversity estimated by CMR Jolly-Seber model (2b). Comparative results for Fig 2a using different minimum species numbers and a separate comparison with fitted splines are presented in supplementary materials. Both methods give similar results (figs. S2 and S3). The crossover estimate is approximate, ~75 to 85 Myr, and we indicate that in the revised MS.

Table 2 needs some additional formatting to make it easier to read - you don't need to use μ , dt , σ in the table and can replace with the words from the legend, and do also explain what aspects of Grainger causality like "characteristic time" mean for those of us less familiar with the method (the description in the methods section is great; thank you!)

[line 333-336] We feel that using the symbols, which correspond directly to what are described in the methods as being estimated, is less confusing for the general reader. We cannot remove the verbal descriptors of each parameter as they pertain to different time series in the same model (e.g., cyclostome.extinction and cheilostome.extinction both have "mu" values in the same model). But we do convert "characteristic time" to half-life, which is easier to understand and we insert an extra line in methods to state how the reparameterization (conversion) is done to be explicit.

Finally, the pyRate reference is almost apologetic. I don't see how it adds much given you use the range-through numbers and the figure is only in the supplement. Either "go big" and use the PyRate uncertainties in the same panel as the range-through numbers or "go home" and drop it. [line 171,173,189,260,280,393] *As explained above, we have revised the MS with emphasis on CMR in the main text, and have rerun PyRate analyses with 100 iterations as suggested by R2, providing these results with discussion in supplementary material section 2.5. Genus sampling, richness and rate estimates using PyRate. In the main text, we refer to PyRate in several places as a method we've used for comparison with our CMR estimates.*

75 - insert only between while and allowing (if I understood the sentence correctly)
[line 383] *inserted the word "only"*

77 - need a reference for the generification
[line 354] *added reference: Hendricks JR, Saupe EE, Myers CE, Hermsen EJ, Allmon WD. 2014 The generification of the fossil record. Paleobiology 40, 511–528.*

99 - is modelling the right word here? accounting for?
[line 81-83] *Sentence now reads: "Capture-mark-recapture (CMR) methods [3] are employed in our genus-level analyses to model sampling rates and account for incomplete sampling, both absent in prior fossil bryozoan biodiversity studies."*

143 - what does "each one" refer to? Species, location or occurrence?
[line [122] *revised to read: "each horizon is retained as a separate locality entry"*

147 - how many does "some" mean?
The comment was referring to locality age assignments that are constrained to a single geologic stage versus ones that could only be constrained to consecutive stages. We now discuss this point in the last paragraph of supplementary material section 1.3 Geologic age standardization. In the FosLocal database, 459 of 1695 localities (27%) spanned more than one ICS stage while in the Age-Only database, 77 of 308 localities (25%) spanned more than one ICS stage.

162 - cite R version number
[line 139] *added R version 4.1.0*

Referee: 2

Comments to the Author. In this study, the authors explore the long-term co-evolutionary dynamics of two groups at local and global scales (cyclostome and cheilostomes bryozoans). They apply a new model-based approach to an incredible dataset. The dataset is almost unparalleled in terms of quality of sampling and reflects an enormous amount of compilation work. Thus, the paper has the potential to be very impactful.

I have a few comments on the methodology (described below) but my main criticism is that the way the paper is currently written doesn't serve the findings of the study well. *We have extensively revised the structure of the MS and the structure of supplementary material—probably best covered in our initial remarks (at the beginning of this response) and among our individual comments to reviewers' concerns.*

The paper starts out very boldly - the authors introduce the importance of inter-clade interactions and the abstract states “inter-clade dynamics are causally linked to each other”. However, my impression after reading the results (1304-1340) is that the results are much less clear than this statement implies.

Further, the finding that “no relationship between any of the origination and extinction rates and species proportions represented in local assemblages” requires more discussion. The patterns observed at a global scale must to some degree be linked to what’s happening at lower spatial scales. The authors state that this results “defies the notion that diversification rates are driven by local representation” - but couldn’t it be that we simply can’t detect the relationship based on available data?

[line 26-30,294-345,386-463] Our results indicate that time series models of cheilostome and cyclostome genus origination and extinction rates are likely causally linked, but that the linkage is not simple and instead suggests a complex feedback behavior. This is one of the important novel findings we wish to highlight. We have revised the abstract to emphasize more clearly much (but not all) of what we consider new and important in our analyses and results. We shortened the introduction (other than the causality section), moving some potentially distracting elements to the discussion to strengthen this focus for the reader. Revisions also have been made to help clarify the paragraphs in Results that relate to our analyses of correlation and Granger causation among time series models so as to help the reader understand our analyses and results. Our discussion now includes several paragraphs related to sampling concerns, including the possibility that relationships may not be detected using available data.

Overall the role of sampling is underplayed - it isn’t really mentioned until the very short discussion. Sampling can never be truly “global” due to incomplete sampling and their modelling does not take this into account. This leads me to wonder whether biased estimates of global metrics are insufficient to be able to detect a causal relationship between assemblage level and global patterns. To me, this is one of the most interesting aspects of the results - since, if we can not establish a relationship between local and global scale dynamics based on a dataset of this condequity, when would we ever be able to do this?

[line 346-463] The discussion has been extensively revised, and new material added there, to both acknowledge and address these concerns. It may also be relevant that we close with a note that these sorts of problems are nearly always persistent worries for studies of fossil biodiversity (even though many published studies do not concede them).

The structure of the paper is a bit unconventional - it has a very lengthy introduction that describes competitive replacement in detail and relatively small and unsatisfying discussion. The discussion doesn’t really come back to competitive replacement or provide a biological explanation for what could be driving the patterns in cyclostomes and cheilostomes.

[line 356-385,439-449] In addition to shortening the introduction, we revised some of the original text there. Comments relevant to ecological abundance and overgrowth competition are now in the discussion. Given that our actual data and analyses are really about "patterns" of local and global biodiversity and not potential ecological drivers, this comment made us realize that the reorganization was necessary. The relocation of these elements will help to clarify that we intended them as context surrounding our analyses, rather than being the main story themselves.

The concept of Granger is difficult to grasp from the introduction. Some of this might be better suited to the methods and given its importance to the results and that most readers will not be familiar with this, I suggest improving this description - perhaps by moving some of the supp explanation of this to the main text.

[line 84-107,191-232] We have revised and expanded the introductory paragraphs on causality and made revisions to the relevant section of methods. These will provide a bit more context and detail introducing Granger causality and its use in our analyses.

PyRate could also be described in a bit more detail in the main text, to enable the reader to better understand the differences between the two modeling approaches (more comments on this below).

As explained above, we have revised the MS with emphasis on CMR in the main text, and have rerun PyRate analyses with results and discussion in supplementary material section 2.5 "Genus sampling, richness and rate estimates using PyRate"

I suggest the authors revisit the structure of the manuscript, focus more on the role of sampling and on how this may affect the results that they obtain in the study.

[line 23-30,77-107,386-463] We have extensively revised the structure of the MS and the supplementary materials. We edited the abstract and introduction to highlight some of our key analyses and sharpen the focus, moving some elements to the discussion. As noted above, CMR methods are used in in the main text in primary analyses. PyRate is identified more clearly as a secondary approach to estimating richness and diversification rates in our ms; the supplementary materials sections 2.4 "Genus sampling, origination and extinction rate estimation using CMR" and 2.5 "Genus sampling, richness and rate estimates using PyRate" both contain analyses and figures relevant to sampling. Finally, we briefly consider several specific sampling concerns in the expanded discussion.

The analyses behind the study were quite well performed, however, the discussion section does not do justice to the results. Please consider expanding it.

[line 386-463] The abstract, introduction, and text surrounding tables 1 and 2 have been revised to address this concern. In particular, we added significant material to the revised discussion.

I also recommend the authors to provide more details regarding the methods in the main text. If the authors improved the text, I think it would make a more accessible and cohesive manuscript on a very important topic.

[line 84-107,108-232,294-345] The abstract and introduction have both been revised to explain more clearly what the main points of the study are. Methods (especially Granger causation and linear SDEs) that are intended to address statistical causation are now better defined in the introcuction. Revisions and details have been added there, in sentences throughout the methods section, and in results text relevant to Tables 1 and 2 (especially on Bayesian inferences). We tried to strike a balance between swamping the average reader with methodological minutiae in the main text and providing deeper discussions of methods and auxiliary analyses in supplementary material. To that end, the sections of supplementary material have been completely reorganized, clustered into method/result groupings, and are now preceded by a table of contents and a full listing of supplementary figures and tables.

Methods

CMR vs PyRate: The comparison between PyRate versus CMR methods is a bit misleading. From the supplement: “PyRate also estimates origination and extinction rates given that taxa are sampled at least once, but capture recapture models relax this assumption, hence also contributing to larger (but more realistic) uncertainties.” While it is true that CMR relaxes this assumption, the two models can not actually be applied to the same data -- CMR uses sampled-in-bin (presence / absence) data while PyRate uses occurrence data. The assumption that taxa are sampled at least once does not explain the difference in uncertainties. Warnock et al. 2021 compare the birth-death process underlying PyRate to one that assumes incomplete species sampling (the fossilised birth-death process) and show that assuming complete has an adverse impact on accuracy - however, it does not have a discernible impact on uncertainty.

[line 153-190] We edited the main text sections "Global genus richness estimation" and "Genus origination and extinction estimation," and the new supplementary materials section 2.5 "Genus sampling, richness and rate estimates using PyRate" for clarity. We made changes so as to avoid misunderstanding that raw data is treated "exactly" the same by CMR and PyRate—it is not. Exactly as R2 pointed out, PyRate does not assume incomplete sampling in the sense that the system only contains as many taxa as are sampled. What PyRate does is to estimate the true start and end points of taxa, such that Pyrate estimates of genera will be higher than “in-bin” observations of genera. This estimation goes beyond range-through taxon estimation; each taxon could have originated before or gone extinct after that first and last “occurrence” in the fossil record. But any taxon that is not recorded at least once in the data will not be included in the estimation. The situation is unlike the case in CMR, where sampling probability also applies to those taxa that are not at all sampled, as long as taxa in the same time interval have the same probability of being sampled/preserved. That is one reason why we chose not to use PyRate for our main analyses. While the uncertainty estimates of CMR modelling may be wider, they are more likely to include the "true signal" than PyRate estimates, which are more precise but less accurate. We hope our rewriting has helped to clarify these points.

It could be the use of more data (i.e. occurrence data) leads to increased precision but a related issue is that PyRate requires exact fossil ages. Because CMR uses sampled-in-bin data, fossil age uncertainty is already accounted for. The developers of PyRate recommend sampling fossil ages from their known interval of uncertainty, repeating the analyses (e.g. 100 times) and constructing credible intervals based on the entire range of output. Ideally, the authors would apply this approach in this paper.

In the original MS, we ran PyRate with default settings of 10 iterations. In this revision, we have rerun the same model 100 times (i.e. each run with different sampled ages of fossil observations) to get at age uncertainty, as recommended. We discuss the procedures in the new supplementary material section 2.5 "Genus sampling, richness and rate estimates using PyRate." Genus richness estimates are shown in figure S7; origination and extinction rates for cyclostomes and cheilostomes are averaged from posterior MCMC outputs and shown in figures S8 and S9.

There's a bit of ambiguity about how the fossil ages are handled - “the combined successive stages are used to bracket an absolute age estimate” - does this refer to age brackets that span more than one interval? Can you clarify how ages that do only span one geological stage are handled? The authors mentioned that they relied on the stratigraphic placement given by the

authors but this would still span a range, even if smaller. How were absolute ages calculated in this case? In addition, assigning absolute ages may introduce additional uncertainties in the analyses, how were these considered? The timescale used in the study also isn't specified in the main text. Perhaps you could add a table with age / stage info to the supp.

In supplementary material section 1.3 "Geologic age standardization," we added table S1 "Geologic stage boundaries and durations." We also revised the relevant paragraph there: "Most localities or geographically circumscribed occurrences of taxa in our data can be constrained to a single ICS geologic stage. We use the chronologic age range from the start to the end of that stage for each such occurrence. In general, the midpoint of the chronologic age range bracketing each occurrence is used in plots and calculations. Where the original author(s) instead assigned the stratigraphic level of a locality to consecutive stages (e.g., Langhian - Serravallian), we constrain the chronological age bracketing that locality from the start of the oldest stage to the end of the youngest stage (e.g., 15.97-11.62 Myr). In the FosLocal database, 459 of 1695 localities (27%) spanned more than one ICS stage while in the Age-Only database, 77 of 308 localities (25%) spanned more than one ICS stage. Where an original stratigraphic assignment referred to part of a stage (e.g., lower Maastrichtian or upper Maastrichtian), we adopt a convention of dividing the relevant international stage into equal halves to provide a chronological age bracket analogous to these divisions."

I also don't fully understand the following statement: "the uncertainty in origination and extinction rates stem directly from the distribution of genera known to be extant but otherwise unsampled" - how exactly does "the distribution of genera known to be extant but otherwise unsampled" lead to estimates uncertainty? Birth-death processes also lead to estimates of uncertainty (whether or not they account for incomplete sampling).

This paragraph has been revised to clarify differences in sources of uncertainty for CMR and PyRate (see comment above). It now appears at the end of supplementary material section 2.5 "Genus sampling, richness and rate estimates using PyRate."

Finally, the authors note that variation in sampling is not accounted for using CMR (although I believe it can be) but this can be accounted for using PyRate.

We could not find the sentence that R2 is referring to here (we searched our original supplementary material file using the terms variation, accounting, account, CMR and PyRate). It seems contradictory that we would have written a statement in this way. Sampling is accounted for in our CMR estimation.

Spline fitting: The authors explored the impact of different degrees of freedom on the fit of cubic splines to their data but the authors did not mention how they decided which fit was chosen as the best fit. Was it based just on a visual inspection or did the user specifically use a model selection method? The R functions they use actually output information that can be used to select the best model - is there a reason the authors chose not to do this? Or if they did, it would be good to mention this.

[line 135-145,234-254] To provide confidence intervals, we re-ran these analyses using a slightly different nonparametric LOESS regression analysis that facilitates an established method for calculating confidence intervals (Figure 1a). These regressions were run with three different data subsets, varying the minimum number of species (10, 15, and 20) in fossil assemblages (figure S2). We did not see the necessity of comparing fits using a model selection

criterion, as there is no "agreed minimum" number of species per assemblage. We have also now clarified in the text that the crossover chronostratigraphic age estimate is approximate, ~ 75-85 Myr. We move the original cubic spline results to supplementary materials section (figure S3). The resulting patterns are comparable, as is the crossover age estimate.

Code: The code associated with this paper is poorly organised - it would not be straightforward for someone else to use. The authors should tidy it up, create subdirectories for the data, scripts and output as well as include clear instructions for other users (i.e. by adding a README).

We have added a README file with instructions for the code and use of relevant files in the Dryad repository. We are unsure why R2 is asking us to make subdirectories, but we assume it is to help the future user. To this end, we identify each of the files by name and indicate what each one contains as we do not see how subdirectories are useful in our case. Each file can also be found in the Rmarkdown file which executes R code for the main analyses.

A private link to the README file, associated code and data are available at Dryad <https://datadryad.org/stash/share/2jQiHPNlwh9ZJsvdQdKJTCzTiIZy4o1Vp2REXbt8FOA> and will be made publically available if and when this ms is accepted and published.

Figures

Suggest adding the official geological time scale to the figures to the plots so it's easier to switch back and forth between the figures and the text.

[line 242,267,285] We added names for several relevant Mesozoic geologic stages, names for several relevant Cenozoic epochs (mentioned in text), chronologic stage or epoch boundaries in main text figures to aid the reader. Chronologic ages have been retained as primary axes in all relevant figures in both the main text and supplementary material. We now list chronologic ages in millions of years for stage boundaries as well as the durations of stage durations in Table S1 of the supplementary material.

Minor

L94, "far surpasses species level documentation for other examples of clade displacement." - Please mention some examples

*[line 77] added references: Briggs JC. 1998 Biotic replacements: Extinction or clade interaction? *BioScience* **48**, 389–395. (doi:10.2307/1313378); Lupia R, Lidgard S, Crane PR. 1999 Comparing palynological abundance and diversity: implications for biotic replacement during the Cretaceous angiosperm radiation. *Paleobiology* **25**, 305–340.*

*(doi:10.1017/S009483730002131X); Silvestro D, Antonelli A, Salamin N, Quental TB. 2015 The role of clade competition in the diversification of North American canids. *PNAS* **112**, 8684–8689. (doi:10.1073/pnas.1502803112)*

1120, "The possibility of multiple realization" - I really have no idea what this means. This whole paragraph is difficult to follow.

[line 356-373] We revised the latter part of the introductory section to sharpen the focus on our primary analyses. The former text considering "multiple realization" is now moved to the discussion and also revised.

136, "facile assertions" - this seems a bit patronizing - the models in this paper also make simplifying assumptions!

[line 38] deleted the word "facile"

1177, I wouldn't say naturally because this has an ambiguous statistical meaning - perhaps say "in a straightforward way".

[line 156-157] revised the sentence to read: Additionally, confidence intervals for genus richness cannot be generated in a straightforward way when using range-through tabulations.

1193, "in a wholly different manner" - this is very vague and I would actually say that the approaches are very similar. They both assume a Poisson process for origination, extinction and sampling, for instance.

[line 185-190] They are not, as explained above. The key difference is that PyRate assumes that all genera are sampled at least once; this is not the case for open models in CMR. This paragraph has been revised as part of our relocation of PyRate comparisons to supplementary material. However, we do remove the wording "in a wholly different manner" and provide a more explicit explanation in the revised supplementary material section 2.5 "Genus sampling, richness and rate estimates using PyRate."

1244, 245, should the number of time series be the same?

[line 244-250] The "four" refer to the 2 extinction and 2 origination time series and the "five" in the next sentence in the original text refers to the 2 + 2 and the of changing proportions of cyclostomes. We modified the text to so there is no confusion.

1286, "For many Cretaceous and Cenozoic stages, higher cheilostome origination rates contrast sharply with cyclostome rates" - what about the large spike at the end?

[line 278-281] We have left this as-is, since the statement is accurate as written. What we are tracking are the general patterns abstracted using the data and methods described. For the most part, we prefer to be circumspect about causal inferences for stage-specific occurrences (seen as outliers).

1304, "Surprisingly, cheilostome origination rates have no detectable relationship to cyclostome origination or extinction rates" is this result surprising, given the patterns in Fig 4a?

[line 298-300] This is in reference to our initial hypotheses. The sentence now reads, "However, cheilostome origination rates have decidedly poorer detectable relationships to cyclostome origination or extinction rates, given the higher weights for the model of no relation between time series (columns 1, 3, model A)."

L320, "Considering the parameter estimates and Bayes factors between models" - be more explicit here.

[line 318-326] Rewritten as "Considering that the mean parameter estimate (Table 2, top portion) $\beta_{\text{cheilostome.extinction.to.cyclostome.extinction}} = 0.522$ (credibility interval of -0.345 to 1.212) is more positive than negative, one acceptable inference is that high cheilostome extinction rates could be related to high extinction rates for cyclostomes, and vice versa. It is plausible that at the same time, high cheilostome extinction rates also induce higher cyclostome origination rates (Table 2 bottom, $\beta_{\text{cheilostome.extinction.to.cyclostome.origination mean parameter estimate}} = 0.275$) and higher origination rates in cyclostomes in turn drive higher

cheilostome extinction rates. However, in the latter comparison model support is lower and β estimates have larger uncertainty bounds (Table 2)."

Appendix C

Responses to referee comments: manuscript RSPB-2021-0637

"When fossil clades "compete": Local dominance, global diversification dynamics, and causation"

Scott Lidgard, Emanuela Di Martino, Kamil Zágorše, and Lee Hsiang Liow

Dear Dr. Hutchinson and referees:

Here is a summary of the minor changes made to the manuscript text. Line numbers listed below refer to this revised version of the manuscript (we show the original line numbers in quoting comments from R2). *Our responses are in italics.*

Referee: 2

Comments to the Author(s).

The authors have done a brilliant job of responding to my questions and suggestions, especially in the introduction, methods and results. Changes to the introduction make it much better suited to the study. The code is also now much more user friendly. (Just for the clarity, the suggestion to create sub-directories is based on the standard programming practice of putting data and scripts in different folders.)

The discussion has also been updated to include more discussion about incomplete sampling. My only outstanding criticism on this front is that, while interesting, the contents of the discussion don't currently link very well back to the authors' results. My suggestion is to improve the connection between the arguments & issues presented in the discussion and the results.

We have added sentences to the discussion to try to emphasize the connections in an even more obvious manner, for example:

[351-354] "However, despite our best efforts with a very large data set and an up-to-date modeling approach, links between changes in species taxonomic dominance within fossil assemblages and diversity dynamics estimated for genera in the two clades were not supported."

[361-367] We do have empirical evidence for encrusting cheilostomes' expected superiority in fossil and Recent overgrowth of cyclostomes [22-24], but we lack sufficient quantification of overgrowth interactions between cheilostomes and cyclostomes in our assemblage data to analyze whether overgrowth interactions are detectably linked to diversification rates in our Granger causal analyses. While proportional species richness is an important parameter in its own right, it is only a proxy for relative abundance, and probably even less so for dominance in overgrowth interactions.

(Note, however, that a large part of the discussion is intended to extrapolate the finite scope of this research to large-scale fossil biodiversity studies in general.)

I would also suggest remove the conclusions section because I don't think it's necessary (and especially wouldn't be needed if the discussion focused more on the results). Overall the paper makes a very valuable contribution to this area of research, and a slightly improved discussion would make it even better.

[469-472] We removed the conclusion, adding a final sentence in the discussion section at the close of the text. "Cheilostomes and cyclostomes, like bivalves and brachiopods [3] were not merely "ships that pass in the night" [78], but fuller understanding of the extent to which time series

histories of local species abundances and overgrowth interactions influenced macroevolutionary outcomes awaits future data compilation and analyses."

Minor comments

182 "to model sampling rates" - do you mean to model the origination, extinction and sampling processes?

[81-84] To clarify, sentences now read, "Capture-mark-recapture (CMR) methods [3] are employed in our genus-level analyses to model origination and extinctions simultaneously with sampling rates and hence to account for incomplete sampling, both absent in prior fossil bryozoan biodiversity studies."

1104 "cheilostome genus origination rates would dampen cyclostome origination rates" - do you mean "did" instead of "would"?

[109] Now reads, "... that increased cheilostome genus origination rates dampened cyclostome origination rates or increased their extinction rates

1140 "4.1.0 (R Core Team 2019)" R version 4.1.0 wasn't available in 2019 but came out in 2021
[142] changed to 2021

1401 suggest citing the following: <https://eartharxiv.org/repository/view/2472/>. Recent work by Roger Close also seems relevant here.

*[407] Added references: Close RA, Benson RBJ, Alroy J, Carrano MT, Cleary TJ, Dunne EM, Mannion PD, Uhen MD, Butler RJ. 2020 The apparent exponential radiation of Phanerozoic land vertebrates is an artefact of spatial sampling biases. *Proceedings of the Royal Society B: Biological Sciences* **287**, 20200372. (doi:10.1098/rspb.2020.0372)*

*Raja NB, Dunne EM, Matiwane A, Khan TM, Nätscher PS, Ghilardi AM, Chattopadhyay D. 2021 Colonial history and global economics distort our understanding of deep-time biodiversity. *EarthArXiv*. (doi:10.31223/X5802N)*

From the supplement:

- "Extinction and origination rates from PyRate analyses are not directly comparable with those from the Pradel model. The latter are transition rates from one time interval to the next, while those from PyRate plotted in figs. S8 and S9 are averages within time intervals" - I could be wrong, but I don't think this means they can't be compared, since we typically interpret them in the same way, i.e. they play the same functional role in hypothesis testing and have been compared in simulation studies.

We agree with the reviewer. Of course they can be compared, but the comparison has to be done with the acknowledgement that there are quantitative differences arising from different assumptions, hence our use of the phrase "not directly comparable".

- "The two classes of models also incorporate different assumptions." - please also make it clear that the methods use different data.

Sentence now reads: "The two classes of models also incorporate different assumptions and use different data as input (taxon observations/non-observations in the Pradel model versus specimen-level observations for PyRate)."

- "We do not compare preservation (sampling) rates for the two models as they are very different in both assumptions and structure" - suggest replacing "structure" with "and the underlying data used" (or similar).

Sentence now reads: "We do not compare preservation (sampling) rates for the two models as they are very different in assumptions, data input, and model structure."

- "the uncertainty stems from the estimation of start and end points of taxon lifespans instead" - and presumably the stochasticity of the underlying birth-death process?

Sentence now reads: "In PyRate, however, the uncertainty stems from the estimation of start and end points of taxon lifespans and the stochasticity of the underlying birth-death process."